# METAXLR - MIXED LANGUAGE META REPRESENTATION TRANSFORMATION FOR LOW-RESOURCE CROSS-LINGUAL LEARNING BASED ON MULTI-ARMED BANDIT

**Liat Bezalel, Eyal Orgad***
Department of Computer Science
Tel-Aviv University, Israel
{liatbezalel, eyalorgad}@mail.tau.ac.il

## ABSTRACT

Transfer learning for extremely low-resource languages is a challenging task as there is no large-scale monolingual corpora for pre-training or sufficient annotated data for fine-tuning. We follow the work of Xia et al. (2021) which suggests using meta learning for transfer learning from a single source language to an extremely low resource one. We propose an enhanced approach which uses multiple source languages chosen in a data-driven manner. In addition, we introduce a sample selection strategy for utilizing the languages in training by using a multi armed bandit algorithm. Using both of these improvements we managed to achieve state-of-the-art results on the NER task for the extremely low resource languages while using the same amount of data, making the representations better generalized. Also, due to the method's ability to use multiple languages it allows the framework to use much larger amounts of data, while still having superior results over the former MetaXL method even with the same amounts of data.

## 1 INTRODUCTION

Multilingual pre-training such as XLM-R Conneau et al. (2020) have presented great results on various NLP tasks for many languages. But in order to achieve that they require a large scale of monolingual data. Unfortunately, this is not the case for extremely low resource languages such Quechua or Ilocano where data for these languages barely exist. Therefore, in the case of a low resource language a way-to-go approach would be using transfer learning. We follow the work of Xia et al. (2021) which uses a meta learning approach. In their work they used a high resource language to learn the task while learning concurrently how to convert representations to the low resource language.

However, the mentioned approach uses only one source language - missing the generalization that could be given using multiple languages. But moving to multi-language approach is not straight forward as different languages can have different effects on the learning process. Selecting languages can be done manually but it is a tedious process that requires linguistic knowledge that sometimes is not widely available. To overcome that, we suggest an approach which uses multiple languages, that are selected in a data driven manner and are balanced during training using a MAB algorithm.

In this paper we show how using multiple languages is more powerful than using a single source language even with the same amount of data. In addition, we propose utilizing multi armed bandit as a sampling strategy to balance the contribution of each language to the training process. Combining both we were able to achieve improved results on the downstream task of NER evaluating languages that are never seen by the pretrained model before. In addition, the language selection is easy and can be done seamlessly.

---

*Both authors contributed equally

| Source / Target | qu | ilo | mhr | mi | tk | gn | Average |
|---|---|---|---|---|---|---|---|
| 1. English 5k (Xia et al., 2021) | 68.67 | 77.57 | 68.16 | 88.56 | 66.99 | 69.37 | 73.22 |
| 2. English 20k (Xia et al., 2021) | 73.04 | 85.99 | 70.97 | 89.21 | 66.02 | 73.39 | 76.44 |
| 3. Related language 5k (Xia et al., 2021) | 77.06 | 75.93 | 69.33 | 86.46 | **73.15** | 71.96 | 75.65 |
| 4. Related language (6k-8k) | 76.47 | 82.3 | 73.78 | **93.53** | 71.07 | 74.07 | 78.54 |
| 5. Uniform selection | 76.27 | 86.41 | 71.43 | 92.67 | **72.9** | **79.65** | 79.88 |
| 6. MetaXLR (ours) | **78.76** | **86.96** | **74.65** | 92.67 | **73.08** | 79.44 | **80.93** |

Table 1: F1 for NER across six settings: (1) Source data size of 5k, English data source only. (2) Source data size of 20k, English data source only. (3) One source language, data size of 5k. (4) MetaXL related source language using the exact same data size as we used in our method (varies between 6k-8k as in Table 2). (5) Choosing languages, uniform distribution. (6) Our method: Source languages defined in Table 2 with MetaXLR algorithm (Algorithm 1).

## 2 METHOD

**Using multiple source languages** To select the source languages we took advantage of both LangRank (Lin et al., 2019) and the languages clusters present in Chiang et al. (2022). First, given a target language $t$ we chose a closely related source language $s_1$ used in Xia et al. (2021) using LangRank. Next, we used the language clusters and mapped $s_1$ 's cluster $c$. Then, we chose $n-1$ arbitrary languages from $c$.

**Multi armed bandit as sampling strategy** Since the languages from the previous step are selected from a large cluster, they may have different effects on the training process as they are yet varied. Thus, we balance the training process by defining a sampling distribution for the source languages while training.

In our strategy we increase the weight of languages that are harder to learn from. The intuition is that in order to generalize the representations properly across the different languages, we should train more with languages that the model struggles with. For making this weighting strategy adaptive to different languages without manual interference, we reduced the problem to a MAB problem where we consider each source language as an arm. Every training step, we select one language from the language distribution and get a reward, which in this case is the loss - The higher it gets, the more the model struggles with this language. The multi armed bandit we used is EXP3 described in Auer et al. (2002). We used it as part of the meta learning algorithm as can be seen in Algorithm 1.

## 3 EXPERIMENTS

Similarly to Xia et al. (2021) we used XLM-R. The data that is used for our experiments is WikiAnn, which contains 282 different languages on the NER task. Results presented in Table 1. Our method outperforms the baselines (1-4) by at least 2.4 F1 score in average, using the same amount of data. Comparing method (1) and (2) emphasizes the importance of the data size. We can also observe the importance of selecting related languages by comparing method (1) and (3). Our method leverages these two observations, by the fact that we are able to use related languages with limited data, and have a large amount of data size combining several related languages together. MetaXL (Xia et al., 2021) works well and improved even by using uniform language selection distribution, by presenting an improved result of at least 1.3 F1 in average. Our language selection algorithm further improves the result by 1.1 F1 score in average. Comparing the two methods (5) and (6), the language selection algorithm performs at least as well as the uniform selection and sometimes outperforms.

## 4 CONCLUSION

In this paper, we study cross-lingual transfer learning for extremely low-resource languages. We broadened MetaXL (Xia et al., 2021), enabling it to use a set of source languages, while choosing from them in the training loop using a Multi Armed Bandit algorithm. We managed to both improve on the results of previous works while simultaneously increasing the pool of usable data and achieve state-of-the-art results for the extremely low resource languages.

URM STATEMENT

The authors acknowledge that at least one key author of this work meets the URM criteria of ICLR 2023 Tiny Papers Track.

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

## A  METAXLR ALOGRITHM

---
**Algorithm 1** Training procedure for MetaXLR
---
**Input:** Input data from the target language data $D_t$ and the source related languages $D_{s_1}...D_{s_K}$
Initialize base model parameters $\theta$ with pretrained XLM-R weights
Initialize parameters of the representation transformation network $\phi$ randomly
Initialize weights $w_0(0) = ... = w_0(K) = 1$
Initialize t = 0

**while** not converged **do**

    Define $p_t(i) = (1 - \gamma)\frac{w_t(i)}{\sum_{j=1}^K} + \frac{\gamma}{K}$

    Sample a source language $s_i$ from $p_t(1)...p_t(K)$

    Sample a source batch $(x_{s_i}, y_{s_i})$ from $D_{s_i}$ and a target batch $(x_t, y_t)$ from $D_t$

    $\theta^{(t+1)} = \theta^{(t)} - \alpha\nabla_\theta\mathcal{L}(f(x_{s_i}; \theta^{(t)}, \phi^{(t)}), y_s)$

    $\phi^{(t+1)} = \phi^{(t)} - \beta\nabla_\phi\mathcal{L}(f(x_t; \theta^{(t+1)}), y_t)$

    $r_t = \frac{\mathcal{L}(f(x_t; \theta^{(t+1)}), y_t)}{p_t(i)}$

    $w_{t+1}(i) = w_t(i)e^{\frac{\gamma \cdot r_t}{K}}$

    $t = t + 1$

**end while**

---

The highlighted lines are the main differences between MetaXL and MetaXLR.

## B  HYPER-PARAMETERS

We used $\gamma = 0.01$ and 12.5k training steps, with batch size of 4. The rest of the hyper-parameters remained the same as in MetaXL (Xia et al., 2021).

## C  RELATED LANGUAGES CLUSTERS

| Target Language | Related language in MetaXL | Related languages in our method |
|---|---|---|
| qu | es | es, pt, it, de, en, ar, he, fr |
| il | id | id, he, ar, de, fr, vi, en |
| mhr | ru | ru, he, ar, de, it, fr, ro, en |
| mi | id | id, he, ar, de, vi, en |
| tk | tr | tr, az, be, uk, sk, lt, sr, cs |
| gn | es | es, pt, it, de, en, ar, he, fr |

Table 2: Target and source languages information on the NER task. The source data size per source language is 1k and the target language data size is 100

## D  REWARD STRATEGIES

| Meta loss as Loss | Uniform Selection | Meta loss as Reward |
|---|---|---|
| 84.06 | 86.41 | **86.96** |

Table 3: F1 score for different reward strategies for the Ilocano target language. Using the loss as a positive or a negative reward affects the performance - rewarding the languages with the high loss, i.e the hard source languages is superior.

## E   CODE

Our code is available at `https://github.com/LiatB282/MetaXLR`

