# OpenReview forum: "MetaXLR - Mixed Language Meta Representation Transformation for Low-resource Cross-lingual Learning based on Multi-Armed Bandit"
_ICLR.cc/2023/TinyPapers — Submitted to Tiny Papers @ ICLR 2023_

### Official Review · Reviewer_HSEe · 2023-03-19

**Confidence:** 5

**Summary Of Contributions:**

Following the work of Xia et al., 2021 an enhanced approach is proposed which uses multiple source languages chosen in a data-driven manner. A sample selection strategy for utilizing the languages in training by using a multi armed bandit algorithm is also introduced.

**Rating:**

High Impact (HI): a submission which meets the reviewing criteria and is predicted to make an impact on the field

**Strengths And Weaknesses:**

Very clearly written and presented paper. The findings are clear, and the basic requirements of the “Tiny” paper are also met. It’s a new perspective on existing work, and as such, the differences are clearly stated.

**Suggested Changes:**

Only one main suggestion: Please read the instructions on Citations within the text.

---

> ### Author Response · Authors · 2023-05-31
> **Authors response**
>
> We thank the reviewer for the thorough feedback and fixed the citations as mentioned.

---

### Official Review · Reviewer_Fswq · 2023-04-02

**Confidence:** 4

**Summary Of Contributions:**

The paper proposes an enhanced approach for transfer learning in extremely low-resource languages by using multiple source languages chosen in a data-driven manner and a sample selection strategy utilizing the languages in training through a multi-armed bandit algorithm.

**Rating:**

High Impact (HI): a submission which meets the reviewing criteria and is predicted to make an impact on the field

**Strengths And Weaknesses:**

Strengths.
- New methodology: The paper suggests a new technique for transfer learning in languages with a very low level of resources that makes use of numerous source languages and a multi-armed bandit algorithm for sample selection.
- State-of-the-art outcomes: The suggested method outperformed earlier approaches by achieving state-of-the-art outcomes on the named entity recognition test for low-resource languages, even with the same quantity of data.
- Data-driven approach: In low-resource environments, choosing source languages and samples based on data-driven criteria enables more effective use of scarce resources.

Weaknesses.

- Lack of analysis: While presenting state-of-the-art results, the authors fail to offer a thorough justification for why their method is superior to earlier ones.
- Reproducibility: The paper presents its findings in a  clear and correct manner. However, it might be hard for a reader to independently produce results


**Suggested Changes:**


- Adding additional analysis: Readers can gain a better understanding of how this new method operates and what makes it effective by reading a thorough examination of why their approach surpasses earlier approaches.
- Increasing the repeatability of tests: To make it simple for others to replicate their work, authors might include detailed descriptions of the datasets utilized during research in addition to code samples.

---

> ### Author Response · Authors · 2023-05-31
> **Authors response**
>
> We thank the reviewer for the thorough feedback and suggestions. We address each of the comments:
>
> 1. Since we have a 2 pages limit, we are limited with the analysis we can provide. In the appendix, section D - you can view 2 different reward strategies for our sampling strategy. There it can be seen how using the loss as a positive or a negative reward affects the performance, This strengthen the fact that the MAB approach can have an impact of the overall performance and can be used to improve the training.
>
> 2. We added to the appendix, in section E, a link to our github code. There, you can find code, scripts and explanation of how to download the dataset and reproduce our result.

---

### Meta-Review · Area_Chair_4e3j · 2023-04-06

**Recommendation:** Invite to present (notable)
**Confidence:** 5

**Metareview:**

An enhanced approach for transfer learning in extremely low-resource languages is presented. It uses multiple source languages chosen in a data-driven manner. A sample selection strategy for utilizing the languages in training using a multi-armed bandit algorithm is introduced.

The proposed methodology, data-driven approach and competitive results in low-resource settings are collectively the strength of this paper.

Maybe a more clear explanation, in Appendix, for reproducibility will be nice to have. Fix the citations within the text according to the instructions for the “Tiny” paper.


**Summary:**

An enhanced approach for transfer learning in extremely low-resource languages is presented. It uses multiple source languages chosen in a data-driven manner. A sample selection strategy for utilizing the languages in training using a multi-armed bandit algorithm is introduced.

**Comments And Feedback To The Authors:**

n/a

**Reason For Not Giving A Higher Recommendation:**

N/A

**Reason For Not Giving A Lower Recommendation:**

Excellent paper with minimal revision.

---

> ### Author Response · Authors · 2023-05-31
> **Authors response**
>
> We thank the reviewer for the thorough feedback and for the decision. For reproducibility - We added a link to our github code in the appendix, provided with instructions and an example. The citations are fixed as well.

---

### Decision · Program_Chairs · 2023-04-07

Invite to present (notable)